# Development and Validation of a QuEChERS-Based LC–MS/MS Method for Natamycin in Imported Agricultural Commodities in Korea

**DOI:** 10.3390/foods14213636

**Published:** 2025-10-24

**Authors:** Ga-Eul-Hae An, Joon-Kyung Oh, Jae-Hyeong Kim, Hee-Ra Chang

**Affiliations:** Department of Pharmaceutical Engineering, Graduate School, Hoseo University, Asan 31499, Republic of Korea; nofallsun1004@naver.com (G.-E.-H.A.); 555wnsrud@naver.com (J.-K.O.); jaebro99@naver.com (J.-H.K.)

**Keywords:** food safety, matrix effects, pesticides, PLS, residues

## Abstract

Natamycin is widely used in other countries for the postharvest treatment of agricultural commodities to prevent fungal growth. However, since no MRL has been set in Korea, natamycin residues are regulated under the Positive List System (PLS) with a uniform limit of 0.01 mg/kg, requiring the development of highly sensitive and reliable analytical methods. In this study, a QuEChERS-based analytical method was developed and validated for the quantification of natamycin in five agricultural commodities—soybean, mandarin, hulled rice, green pepper, and potato—using liquid chromatography–tandem mass spectrometry (LC-MS/MS). Extraction using methanol with 3 g of MgSO_4_ resulted in high recoveries without crystallization, and clean-up with MgSO_4_ and C18 effectively reduced matrix interferences blow <50%. Natamycin was detected in all five matrices at 6.8 min without any interfering peaks. The MLOQ was determined at 0.01 mg/kg for all five matrices. The mean recoveries (82.2–115.4%) and %CV values (1.1–4.6%) values were within the acceptance criteria defined by the CODEX guidelines. Matrix effects were classified as “soft” for mandarin (|ME| < 20%) and “medium” for soybean, hulled rice, green pepper, and potato (20% ≤ |ME| < 50%). The analytical method for natamycin was validated as suitable for regulatory safety monitoring under the Korean PLS.

## 1. Introduction

Natamycin is a polyene macrolide fungicide produced by Streptomyces natalensis, which inhibits fungal spore germination by binding to ergosterol in the microbial cell membrane [1,2,3,4]. Natamycin is widely used in processed foods such as cheese, yogurt, juice, and wine to inhibit spoilage microorganisms [5,6]. In addition to its application in processed foods, it is also used postharvest as a pesticide on agricultural commodities such as citrus fruits, pineapples, and mushrooms, typically through spraying or surface coating, to control fungal contamination and maintain product quality during storage [7]. Reflecting this use, the antifungal efficacy of natamycin as a postharvest fungicide has been extensively studied in processed foods and certain fresh produce such as blueberries, grapes, and cherry tomatoes [8,9]. However, most previous studies have focused on beverages, dairy products, processed foods, or wine, whereas research on natamycin residue levels in raw agricultural produce continues to be limited. Consequently, it is difficult to determine whether and to what extent natamycin residues may be present in agricultural commodities. Therefore, from the perspective of agricultural commodity consumption, the presence of pesticide residues may pose potential risks to consumer safety [10,11,12,13,14]. Consequently, many national food regulatory agencies continue to revise guidelines for the registration of natamycin as a pesticide based on toxicological and exposure assessments. For example, the U.S. Environmental Protection Agency (EPA) has assessed the tolerances of natamycin for citrus fruits, avocados, and mangoes [15]. However, natamycin has not yet been registered as a pesticide in Korea, and consequently, no MRL has been established. As a result, under the Positive List System (PLS) enforced by the Ministry of Food and Drug Safety (MFDS), a default residue limit of 0.01 mg/kg is applied to unregistered pesticides, including natamycin [16,17]. For the safety control of natamycin in imported agricultural commodities, highly sensitive and accurate analytical methods are required to detect residues at concentrations below 0.01 mg/kg. Natamycin is an amphiphilic compound containing a lactone ring with non-polar conjugated double bonds and several polar functional groups, such as hydroxyl groups [18,19]. These structural characteristics result in poor solubility in water but high solubility in polar organic solvents, including methanol and acetonitrile [20,21,22]. In addition, the polyene structure is highly sensitive to ultraviolet (UV) light, leading to degradation with increased exposure time, and exhibits rapid degradation under both low and high pH conditions [13,23,24]. Considering these characteristics, an accurate and precise analytical method for detecting natamycin in agricultural commodities is required.

In recent years, the QuEChERS (Quick, Easy, Cheap, Effective, Rugged, and Safe) method, coupled with liquid chromatography-tandem mass spectrometry (LC-MS/MS), has been widely applied for the determination of pesticide residues in various agricultural commodities [25,26]. The QuEChERS method, developed by Anastassiades in 2003, provides a rapid and simplified alternative to conventional liquid–liquid extraction (LLE) and solid-phase extraction (SPE) [27]. However, attributable to the complex nature of agricultural matrices, non-target compounds may interfere with analytical accuracy, requiring the application of highly selective and sensitive detection techniques such as LC-MS/MS [28]. This study developed and validated a QuEChERS-based analytical method employing LC-MS/MS for the determination of natamycin in imported agricultural commodities. Thereafter, the validated method was applied to monitoring imported agricultural products, and the detected levels were evaluated to ensure compliance with regulatory limits for consumer safety.

## 2. Materials and Methods

### 2.1. Chemicals and Reagents

A natamycin standard (purity 91.13%) was purchased from HPC Standards GmbH (Cunnersdorf, Germany). The physicochemical properties of natamycin are summarized in Table A1. HPLC grade organic solvent-including methanol, and water-were purchased from J.T.Baker^®^ (Avantor Performance Materials Korea Ltd., Suwon-Si, Republic of Korea). Extraction reagents for the QuEChERS method were obtained from Agilent Technologies Korea Ltd. (Seoul, Republic of Korea). The AOAC method kit (No. 7982-7755) contained 6 g anhydrous magnesium sulfate (MgSO_4_) and 1.5 g sodium acetate (NaOAc); the EN method kit (No. 5982-7650) contained 4 g MgSO_4_, 1 g sodium chloride (NaCl), 1 g trisodium citrate dihydrate (Na_3_Cit·2H_2_O), and 0.5 g disodium hydrogencitrate sesquihydrate (Na_2_HCit·1.5H_2_O); and the Original method kit (No. 5982-7550) contained 4 g MgSO_4_ and 1 g NaCl. Dispersive solid-phase extraction (d-SPE) sorbents, including MgSO_4_, octadecylsilane (C18), and graphitized carbon black (GCB), were also purchased from Agilent Technologies Korea Ltd. (Seoul, Republic of Korea). Formic acid (HPLC grade, purity 99%), used for acidifying the mobile phase, was supplied by FUJIFILM Wako Pure Chemical Corporation (Osaka, Japan).

### 2.2. LC-MS/MS Instrumental Conditions

Natamycin analysis was performed using a liquid chromatography–tandem mass spectrometry (LC-MS/MS) system comprising an AB SCIEX QTRAP 5500 mass spectrometer (AB SCIEX, Concord, ON, Canada) coupled with an AB SCIEX ExionLC™ Series UHPLC (AB SCIEX, Concord, ON, Canada). Chromatographic separation was carried out on a reversed-phase Unison UK-C18 column (100 mm × 2.0 mm, 3 µm, Imtakt, Kyoto, Japan), maintained at 35 °C, with an injection volume of 5 µL. The mobile phases consisted of 0.1% formic acid in water (A) and 0.1% formic acid in methanol (B). Gradient elution was performed at a flow rate of 0.2 mL/min under the following conditions: the initial composition of 50:50 (A:B, *v*/*v*) was held from 0.0 to 4.0 min, followed by a linear increase in B to 100% from 4.0 to 5.0 min. This condition was maintained until 7.0 min to ensure complete elution of natamycin. The mobile phase was then returned to 50:50 (A:B, *v*/*v*) from 8.0 to 12.0 min for column re-equilibration. The total run time was 12 min, and natamycin was detected at a retention time (RT) of 6.8 min.

Mass spectrometric detection was performed in positive electrospray ionization (ESI^+^) mode using multiple reaction monitoring (MRM) to enhance selectivity and sensitivity.

The ion spray voltage was set to +4500 V, and the source temperature was maintained at 550 °C. Nitrogen was used as the collision gas. MRM parameters were optimized by direct infusion of a 20 µg/L natamycin standard solution. The precursor ion was observed at *m*/*z* 666.2, corresponding to the protonated molecular ion [M + H]^+^. Two product ions with the highest signal intensities were selected as the quantifier and qualifier ions. The final MRM conditions, including precursor/product ion transitions, cone voltages, collision energies, and retention time, are summarized in Table A2.

### 2.3. Selection and Preparation of Samples

For the optimization and validation of the analytical method, two commodities—soybean (legume vegetable) and mandarin (fruit)—and five commodities—hulled rice (cereal grain), potato (root and tuber vegetable), soybean (legume vegetable), mandarin (fruit), and green pepper (fruiting vegetable)—were selected, respectively, based on the Codex food classification system and the national priority list of commonly consumed foods. All samples were purchased as pesticide-free organic agricultural products from a certified organic food market.

A total of 102 imported agricultural products were collected from both online and offline markets for natamycin monitoring. The samples consisted of 21 legumes (3 soybeans, 6 kidney beans, 2 lentils, 2 chickpeas, 2 adzuki beans, 2 peas, 2 cowpeas, 1 lupin bean, and 1 fava bean), 32 fruits (4 oranges, 3 cherries, 3 lemons, 2 mangos, 5 bananas, 5 grapes, and 1 each of lime, raspberry, blueberry, melon, kiwi, grapefruit, mangosteen, papaya, avocado, and pineapple), 19 cereals (4 quinoas, 4 kamuts, 4 barleys, 2 farros, 2 oat rices, and 1 each of millet, corn, and calrose rice), 28 vegetables (4 green peppers, 3 broccoli, 2 green onions, 2 garlic scapes, 2 shallots, 2 asparagus, 2 Chinese cabbages, and 1 each of spinach, okra, onion, carrot, radish, pumpkin, jicama, cabbage, burdock, ginger, and red bell pepper), and 2 root crops (2 potatoes). These agricultural products originated from 17 countries: Australia, Belgium, Canada, Chile, China, Israel, Italy, the Netherlands, New Zealand, Peru, the Philippines, the Republic of Costa Rica, the Republic of Madagascar, Spain, Thailand, the United States, and Vietnam. All samples were immediately frozen at −70 °C in a deep freezer for at least 24 h, and then homogenized using a stainless-steel homogenizer to ensure complete sample uniformity. The homogenized cereal and legume samples were further passed through a 420 μm stainless-steel mesh sieve to obtain a uniform particle size. All homogenized samples were transferred into high-density polyethylene (HDPE) amber bottles and stored at −20 °C prior to analysis.

### 2.4. Optimization and Validation of Extraction and Purification Conditions

For method optimization, soybean and mandarin were selected as priority matrices based on their distinct physicochemical characteristics. Soybean is characterized by a high lipid content, while mandarin exhibits high acidity and peels rich in pectin and complex aromatic compounds. These matrix characteristics are known to increase interferences and matrix effects, thereby complicating the analytical process [29,30]. In particular, soybean samples were pre-hydrated to enhance solvent accessibility, as their dried state reduces extraction efficiency when using water-miscible organic solvents [31]. The efficiency of extraction and purification was determined by recovery. For method optimization in this study, the efficiencies of extraction and purification were evaluated using the QuEChERS method. Methanol was selected as an extraction solvent based on the amphiphilic nature of natamycin—containing polar (–OH, –COOH) and nonpolar (–C=C–, –CH_3_) functional groups [11,12,13,14,15]. To optimize the extraction method, the QuEChERS method was compared with the Original method, EN 15662, and AOAC 2007.01 extraction. The clean-up step using dispersive solid-phase extraction (d-SPE) to minimize matrix effects was optimized with various combinations of sorbents, including C18 and graphitized carbon black (GCB).

The method validation parameters were evaluated in accordance with both the CODEX (CAC/GL 40-1993) [32] and the MFDS Guideline of Standard Procedures of Test Methods for Foods and Other Substances (2025) [33]. Validation parameters included specificity, linearity, limits of detection (LOD) and quantification (LOQ), accuracy, precision, and matrix effect. Specificity was assessed by comparing the chromatograms of blank and spiked samples to determine whether any interfering substances had the same retention time and *m*/*z* values as natamycin. The ion ratio was evaluated to confirm identification based on the consistent relative abundance of the quantifier and qualifier ions, and was calculated using data obtained from matrix-matched standard solutions (0.002, 0.005, 0.01, 0.02, 0.05, and 0.06 μg/mL) and recovery samples (0.01 mg/kg, *n* = 5). Linearity was performed by constructing a calibration curve using the same six concentration levels (0.002, 0.005, 0.01, 0.02, 0.05, and 0.06 μg/mL) of matrix-matched standard solutions, and the coefficient of determination (R^2^) was confirmed to meet the acceptance criterion (≥0.98) [34].

The instrumental limit of detection (ILOD) and instrumental limit of quantitation (ILOQ) were determined using natamycin standard solutions at six concentrations (0.002–0.06 μg/mL), each analyzed in seven replicates. The standard deviation of the y-intercept (σ) and the slope (S) from the calibration curve were used to calculate the limits according to the following equations: ILOD = 3.3 × (σ/S) and ILOQ = 10 × (σ/S). The signal-to-noise (S/N) ratios for the ILOD and ILOQ, measured by LC-MS/MS, were confirmed to meet the acceptance criteria of ≥3 and ≥10, respectively. The method limit of quantitation (MLOQ) was calculated as 0.01 mg/kg using the ILOQ and relevant analytical parameters, including the sample weight and sample solution volumes [35].

Accuracy and precision were evaluated at three fortification levels: MLOQ (0.01 mg/kg), 10× MLOQ (0.1 mg/kg), and 50× MLOQ (0.5 mg/kg). Natamycin standard solutions were spiked into blank matrices, and five replicate analyses were performed at each level. Mean recovery (%) and the coefficient of variation (CV, %) were calculated to assess accuracy and repeatability in accordance with the acceptance criteria specified by the MFDS guidelines [33].

Matrix effects on signal intensity were calculated from the ratio of the slopes of calibration curves (0.002, 0.005, 0.01, 0.02, 0.05, and 0.06 μg/mL; five replicates) constructed using standard solutions and matrix-matched solutions. The matrix-matched standard solutions were prepared by mixing the natamycin standard solution with blank sample extract to obtain a solution containing 90% blank extract. Positive values indicated ion enhancement, whereas negative values indicated ion suppression. The absolute matrix effect was classified into three levels: <20% (low/soft), 20% to <50% (moderate/medium), and ≥50% (high/strong) [36].

All data acquisition and processing were performed using Analyst software (Sciex, Version 1.6.3, Framingham, MA, USA) on an LC-MS/MS system equipped with an AB SCIEX QTRAP 5500 mass spectrometer and an ExionLC™ UHPLC. Statistical analyses were conducted using Microsoft Excel 2016 (Microsoft, Redmond, WA, USA).

### 2.5. Applicability of the Established Analytical Method

Monitoring was conducted to determine natamycin residues in imported agricultural products currently distributed in the Korean market. A total of 102 samples were collected from both online and offline sources and classified into five categories: legumes, fruits, cereals, vegetables, and root crops. Natamycin levels were quantified using matrix-matched calibration curves based on the validated analytical method. Finally, the detected levels in imported agricultural products were evaluated for compliance with the Korean Positive List System (PLS) to ensure consumer safety.

## 3. Results and Discussion

### 3.1. Optimization of Extraction and Purification Conditions

Soybean and mandarin were selected as representative matrices due to their complex physicochemical characteristics. Their high lipid content and acidity, respectively, enhanced matrix interferences and complicated residue analysis.

#### 3.1.1. Optimization of Extraction Conditions

Natamycin was extracted with methanol (solubility approximately 3000 mg/L), and recoveries were evaluated using three QuEChERS methods—Original, EN 15662, and AOAC 2007.01 (Table 1)—in accordance with CODEX (CAC/GL 40-1993) and MFDS guidelines [37,38,39]. The recoveries obtained with the EN 15662 method for soybean at 0.01 mg/kg and for mandarin at 0.1 mg/kg were not within the acceptable range of 70–120%. Notably, the mean recovery for soybean at 0.01 mg/kg was 126.8%, which is likely attributable to strong salting-out effects induced by the QuEChERS procedure, resulting in disproportionately higher recoveries compared with other fortification levels [40]. In contrast, the recoveries and CV values obtained with the Original and AOAC 2007.01 methods were within the acceptance criteria for both soybean and mandarin. The Original method generally showed higher and more consistent recoveries and was selected as the optimized extraction method.

The QuEChERS original method includes a salting-out extraction with NaCl; therefore, it is important to remove salts that can precipitate. After extraction using the original method, salt precipitation was observed in the analytical sample solution, which may compromise analytical reproducibility [41]. In the extraction method, 3 g of MgSO_4_ was employed and NaCl was not used to prevent salt precipitation after extraction. The recovery and %CV of natamycin in both soybean and mandarin ranged from 79.0 to 109.2% and 0.3 to 1.3%, respectively, which were within the acceptable criteria specified by the guidelines.

#### 3.1.2. Optimization of Purification Conditions

To reduce nonpolar and pigment-related interferences, purification efficiency was evaluated by adjusting the amounts of C18, MgSO_4_, and GCB. C18 is effective in removing lipids and other nonpolar substances, while GCB enhances the removal of pigments such as carotenoids, chlorophylls, and sterols [42]. For the three purification conditions tested—MgSO_4_ 150 mg + C18 25 mg, MgSO_4_ 900 mg + C18 150 mg, and MgSO_4_ 900 mg + C18 150 mg + GCB 20 mg—natamycin recoveries met the acceptance criteria (60–120% with CV ≤ 32% at 0.01 mg/kg, 70–120% with CV ≤ 22% at 0.1 mg/kg, and 70–110% with CV ≤ 18% at 0.5 mg/kg). However, the addition of GCB at levels above 50 mg resulted in recoveries that fell outside the acceptable range (Table 2). Previous studies have indicated that GCB may lead to analyte loss depending on the molecular structure during the purification step [42].

The matrix effect was evaluated using purification methods (MgSO_4_ 150 mg + C18 25 mg, MgSO_4_ 900 mg + C18 150 mg, and MgSO_4_ 900 mg + C18 150 mg + GCB 20 mg) in which recoveries were within the acceptable range (Figure 1). Within these purification conditions, the method employing MgSO_4_ 900 mg + C18 150 mg resulted in matrix effects classified within the soft or medium range, whereas the other methods exceeded the strong range [43].

Finally, the analytical method for natamycin was optimized as follows: Ten grams of sample (5 g for cereals and legumes, pre-hydrated with 5 mL distilled water for 30 min) were weighed into a 50 mL conical tube. Methanol (20 mL, or 10 mL for cereals and legumes) was added, and the mixture was vigorously shaken for 1 min. Magnesium sulfate (3 g) was then added, followed by another 1 min of shaking, and the mixture was centrifuged at 4000 rpm for 5 min. A 5 mL aliquot of the supernatant was transferred to a tube containing 900 mg MgSO_4_ and 150 mg C18, shaken for 1 min, and centrifuged again at 4000 rpm for 5 min for purification. The purified extract was filtered through a syringe filter and analyzed by LC–MS/MS.

### 3.2. Method Validation

The optimized method described above was validated for hulled rice (cereal grain), potato (root and tuber vegetable), soybean (legume vegetable), mandarin (fruit), and green pepper (fruiting vegetable) to verify its robustness and reliability.

#### 3.2.1. Specificity and Linearity

Specificity was confirmed by comparing chromatograms of blank samples with those of blank samples spiked with natamycin at the MLOQ (0.01 mg/kg). Natamycin was consistently detected at a retention time of 6.8 min in all five matrices (soybean, mandarin, hulled rice, green pepper, and potato), with no interfering peaks observed at the same retention time and mass-to-charge ratio (*m*/*z*) (Figure A1). The average ion ratios of the matrix-matched standard solutions (0.002–0.06 μg/mL) were below 10% across all matrices: soybean (0.51%), mandarin (0.51%), hulled rice (0.47%), green pepper (0.51%), and potato (0.48%). The relative tolerances were calculated as −3.18% for soybean, −0.81% for mandarin, 2.15% for hulled rice, 3.34% for green pepper, and 0.53% for potato, all of which were within the acceptable tolerance limit (±50%) according to the 2002/657/EC guideline [44]. Linearity was confirmed using matrix-matched standard solutions at six concentration levels (0.002–0.06 μg/mL), with coefficients of determination (R^2^ > 0.99) obtained for matrices (Table A3).

#### 3.2.2. Method Limit of Quantitation (MLOQ)

The ILOD and ILOQ, calculated from the standard deviation of the y-intercept (σ) and the slope (S) of the calibration curve (0.002–0.06 μg/mL, 7 replicates), were 0.002 μg/mL and 0.005 μg/mL, respectively. The signal-to-noise (S/N) ratios of the ILOD and ILOQ measured by LC–MS/MS met the acceptance criteria of ≥3 and ≥10, respectively. The MLOQ was determined for all five matrices by multiplying the calculated ILOQ (0.005 μg/mL) by the sample solution volume (20 mL, or 10 mL for cereals and legumes) and dividing by the sample weight (10 g, or 5 g for cereals and legumes), resulting in a value of 0.01 mg/kg, which complied with the requirements of the Korean PLS.

#### 3.2.3. Accuracy and Precision

Accuracy and precision were assessed by spiking blank matrices with natamycin at three concentration levels—MLOQ (0.01 mg/kg), 10× MLOQ (0.1 mg/kg), and 50× MLOQ (0.5 mg/kg)—followed by recovery testing in five replicates at each level. The mean recoveries (82.2–115.4%) and %CV values (1.1–4.6%) values were within the acceptance criteria defined by the CODEX guidelines, which specify recovery ranges of 60–120% with CV ≤ 32% at the LOQ, 70–120% with CV ≤ 22% at 10× LOQ, and 70–110% with CV ≤ 18% at 50× LOQ [32] (Table 3). These results demonstrate the high accuracy and analytical reliability of the method.

#### 3.2.4. Matrix Effect

The matrix effects (MEs), based on the slopes of the calibration curves, were −24.4% for soybean, −10.0% for mandarin, 32.2% for hulled rice, −21.5% for green pepper, and 36.5% for potato (Figure 2). The ME for mandarin fell within the “soft” matrix effect range (|ME| < 20%), indicating negligible interference. The MEs for soybean, hulled rice, green pepper, and potato were classified as “medium” matrix effects (20% ≤ |ME| < 50%). Therefore, natamycin was quantified using the matrix-matched calibration curve for accurate determination [45].

### 3.3. Monitoring of Imported Agricultural Products

Monitoring of natamycin was conducted on 102 imported agricultural products, including 21 legumes, 32 fruits, 19 cereals, 28 vegetables, and 2 root crops, originating from 17 countries including Australia, China, Spain, the United States, and Vietnam. Natamycin was not detected in any of the 102 imported agricultural products. This result is attributable to the photo-degradation characteristics of natamycin, which is known to degrade upon exposure to light sources such as ultraviolet (UV), LED, and fluorescent lighting during distribution and retail processes associated with importation [46]. In a previous study, natamycin was not detected in wine due to degradation during the production process [47]. The absence of detectable residues indicates compliance with Korea’s PLS (≤0.01 mg/kg) and provides reassurance regarding consumer safety. Although this study was limited to 102 samples, the findings provide meaningful evidence on the residue status of natamycin in agricultural commodities. The 2017 JMPR evaluation noted that the lack of sufficient residue data, particularly for commodities such as citrus fruits, pineapples, and mushrooms, makes it difficult to establish reliable MRLs [48]. The present study provides a scientifically validated method and initial monitoring data that may serve as a basis for addressing this gap. Future studies incorporating a wider range of samples and expanded stability evaluations, building on the monitoring results of this study, will further strengthen the evidence base and support more comprehensive consumer protection.

## 4. Conclusions

Natamycin, although widely used in agricultural practices in other countries, is not registered for use in Korea. Therefore, an analytical method is required to manage imported agricultural products at the enforcement level. In this study, a QuEChERS-based analytical method using LC–MS/MS was developed and validated for the determination of natamycin residues in agricultural commodities. Method validation was performed for five representative crops—soybean, mandarin, hulled rice, green pepper, and potato—in compliance with CODEX guidelines (CAC/GL 40-1993) and the standard testing protocol issued by the Ministry of Food and Drug Safety (MFDS) [34] confirming the reliability of the method. The application of the validated method to 102 imported agricultural products from 17 countries revealed no detectable residues of natamycin. These results were within the default tolerance level (≤0.01 mg/kg) established under Korea’s PLS for regulating agricultural products. Although this study was limited to 102 samples, it provides meaningful evidence regarding the status of natamycin residues in agricultural commodities. In particular, given the limited international monitoring data for commodities such as citrus fruits, pineapples, and mushrooms, which makes establishing reliable maximum residue limits (MRLs) challenging, these findings offer important evidence on the occurrence of natamycin residues in agricultural commodities. Building on these findings, future studies incorporating a broader range of samples and extended stability evaluations will further reinforce the evidence base and support more comprehensive consumer protection. Thus, the analytical method developed and validated in this study is reliable and reproducible for monitoring natamycin residues under Korea’s PLS and contributes to ensuring consumer safety.

## Figures and Tables

**Figure 1 foods-14-03636-f001:**
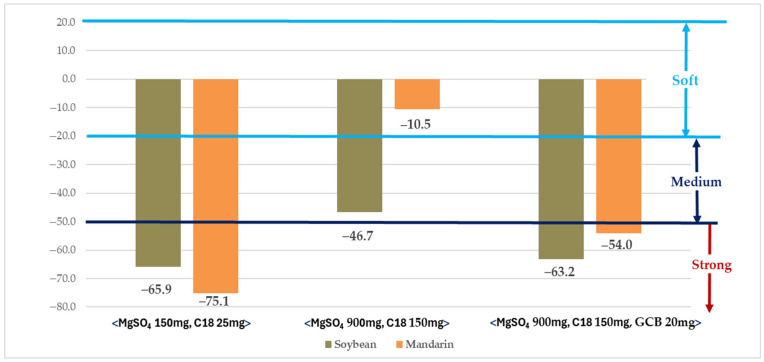
Evaluation of matrix effects under different purification conditions.

**Figure 2 foods-14-03636-f002:**
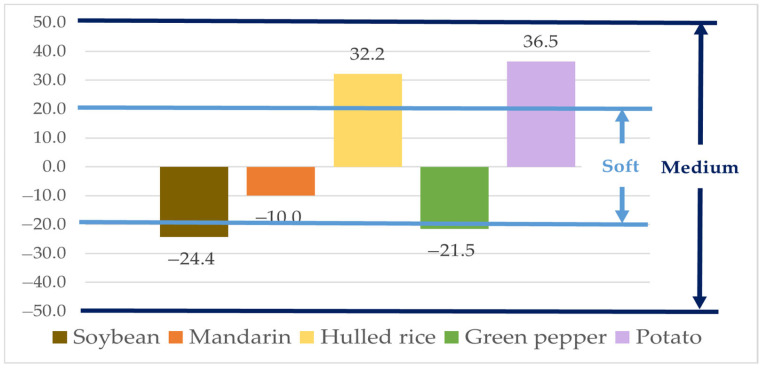
Matrix effect of Natamycin in agricultural matrices.

**Table 1 foods-14-03636-t001:** Recoveries of natamycin in soybean and mandarin using the Original, EN 15662, and AOAC 2007.01 extraction methods.

Extraction Solvent	Extraction Method	Spiked Level (mg/kg)	Recovery (%), *n* = 3			
			Soybean		Mandarin	
			Mean ± SD ^a^	%CV ^b^	Mean ± SD	%CV
Methanol	Original	0.01	88.3 ± 1.7	2.0	93.6 ± 1.4	1.5
		0.1	90.5 ± 3.4	3.7	98.4 ± 0.9	1.0
		0.5	88.6 ± 3.4	3.8	96.8 ± 1.2	1.3
	EN 15662	0.01	126.8 ± 1.2	0.9	73.2 ± 0.4	0.6
		0.1	82.2 ± 0.1	0.2	68.0 ± 0.4	0.6
		0.5	90.4 ± 0.5	0.6	77.5 ± 0.5	0.6
	AOAC 2007.01	0.01	90.5 ± 2.3	2.6	88.9 ± 3.7	4.2
		0.1	76.0 ± 0.2	0.3	92.5 ± 1.4	1.6
		0.5	73.0 ± 0.8	1.1	86.2 ± 0.6	0.7

^a^ Standard deviation; ^b^ Percentage coefficient of variation, calculated as (SD/Mean) × 100.

**Table 2 foods-14-03636-t002:** Recoveries of natamycin in soybean and mandarin under different purification steps using various combinations of MgSO_4_, C18, and GCB.

Purification Condition	Spiked Level (mg/kg)	Recovery (%), *n* = 3			
		Soybean		Mandarin	
		Mean ± SD ^a^	%CV ^b^	Mean ± SD	%CV
MgSO_4_ 150 mg, C18 25 mg	0.01	79.0 ± 0.7	0.9	88.5 ± 0.6	0.7
	0.1	94.2 ± 1.3	1.3	87.4 ± 1.0	1.2
	0.5	109.2 ± 0.4	0.3	109.0 ± 0.9	0.8
MgSO_4_ 900 mg, C18 150 mg	0.01	92.1 ± 0.7	0.8	84.2 ± 2.5	3.0
	0.1	113.5 ± 1.9	1.7	92.6 ± 4.0	4.3
	0.5	105.8 ± 1.0	1.0	91.5 ± 2.9	3.2
MgSO_4_ 900 mg, C18 150 mg, GCB 20 mg	0.01	90.5 ± 1.5	1.7	95.6 ± 1.3	1.3
	0.1	79.7 ± 0.8	1.0	74.2 ± 0.8	1.0
	0.5	83.6 ± 0.8	0.9	79.1 ± 0.4	0.5
MgSO_4_ 900 mg, C18 150 mg, GCB 50 mg	0.01	85.6 ± 0.3	0.4	49.9 ± 2.1	4.1
	0.1	67.7 ± 1.1	1.7	34.3 ± 1.9	5.6
	0.5	74.2 ± 1.1	1.4	47.1 ± 1.4	2.9
MgSO_4_ 900 mg, C18 150 mg, GCB 100 mg	0.01	35.5 ± 0.2	0.5	16.4 ± 0.7	4.3
	0.1	37.1 ± 0.9	2.5	12.9 ± 1.5	11.6
	0.5	45.3 ± 0.7	1.5	13.3 ± 1.0	7.7
MgSO_4_ 900 mg, C18 150 mg, GCB 150 mg	0.01	9.7 ± 0.3	3.1	2.3 ± 0.1	5.3
	0.1	24.2 ± 1.7	7.1	4.7 ± 0.2	5.0
	0.5	32.5 ± 1.1	3.4	6.2 ± 0.3	5.0

^a^ Standard deviation; ^b^ Percentage coefficient of variation, calculated as (SD/Mean) × 100.

**Table 3 foods-14-03636-t003:** Recoveries and %CV values for natamycin in five representative agricultural commodities.

Matrix	Spiked Level (mg/kg)	Recovery (%), *n* = 5	
		Mean ± SD ^a^	%CV ^b^
Soybean	0.01	91.4 ± 1.0	1.1
	0.1	115.4 ± 3.0	2.6
	0.5	104.6 ± 3.7	3.6
Mandarin	0.01	82.2 ± 3.8	4.6
	0.1	90.7 ± 3.9	4.3
	0.5	89.5 ± 3.8	4.3
Hulled rice	0.01	91.4 ± 2.2	2.4
	0.1	86.1 ± 1.6	1.9
	0.5	84.4 ± 1.9	2.2
Green pepper	0.01	90.2 ± 1.5	1.7
	0.1	91.6 ± 2.4	2.6
	0.5	88.5 ± 2.6	2.9
Potato	0.01	99.4 ± 2.1	2.1
	0.1	88.8 ± 3.9	4.4
	0.5	90.3 ± 1.9	2.2

^a^ Standard deviation; ^b^ Percentage coefficient of variation, calculated as (SD/Mean) × 100.

## Data Availability

The original contributions presented in this study are included in the article. Further inquiries can be directed to the corresponding author.

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
