# Peer review of "Development and Validation of a QuEChERS-Based LC–MS/MS Method for Natamycin in Imported Agricultural Commodities in Korea"

_foods, 2025, doi:10.3390/foods14213636_

Round 1

Reviewer 1 Report

Comments and Suggestions for Authors

Rationale for Recommendation:The manuscript presents a QuEChERS-LC-MS/MS method for natamycin determination in agricultural products. While the work is competent technically, it lacks sufficient novelty and scientific impact. The approach is a standard application of well-established techniques without introducing new concepts, significant performance improvements, or innovative solutions to existing challenges. The monitoring results, showing no detections in 102 samples, provide limited scientific value for a research journal.

Detailed Comments and Questions

  1. Novelty and Scientific Contribution (1-4)

Innovation Deficit: What is the core methodological innovation of this work? How does it substantively improve upon existing QuEChERS-LC-MS/MS methods for natamycin in terms of sensitivity, throughput, cost, or ruggedness?

Study Design: Natamycin's photosensitivity and pH lability are mentioned. How were light and pH rigorously controlled during sample preparation and analysis to prevent degradation? Specific quality control measures are not described.

Limited Practical Value: The finding of no natamycin in 102 samples, while useful for surveillance, offers weak scientific evidence. Did the authors consider analyzing certified reference materials or spiking real samples to better demonstrate the method's efficacy for enforcement?

Method Generality: Validation was limited to five matrices. How would the method perform with high-lipid (e.g., nuts) or highly pigmented (e.g., paprika) matrices? Please provide an assessment or justification.

  1. Experimental Methods and Details (5-12)

Sample Pre-treatment: For "pre-hydration" of cereals/legumes, what is the rationale for adding exactly 5 mL of distilled water? How does this affect recovery and matrix effects?

Critical Reagent Choice: The authors chose 3g MgSOâ‚„ without NaCl to avoid precipitation. How was the trade-off between reduced salting-out efficiency and the prevention of precipitation evaluated and justified?

Clean-up Strategy: GCB > 50 mg reduced recoveries. For pigment-rich samples, was a combination of sorbents (e.g., PSA with low GCB) considered to balance cleanup and recovery?

Matrix Effect Assessment: Medium matrix effects were reported. Please provide the raw slope data from solvent-based vs. matrix-matched calibrations to allow readers to independently assess the magnitude of these effects.

Validation Completeness: Robustness and repeatability testing are missing. How do variations in extraction time, centrifugation speed, or flow rate affect results?

Instrumental Parameters: Given natamycin's polarity, could the gradient be adjusted (e.g., higher initial aqueous phase) to elute the analyte earlier and increase throughput?

Standard Purity Correction: The standard purity is 91.13%. Were all quantitative calculations corrected for this purity?

Data Availability: "Data available on request" is insufficient. Please provide key raw data (e.g., MRM chromatograms, calibration curves) as supplementary materials.

  1. Results Analysis and Discussion (13-16)

Insufficient Mechanistic Discussion: The discussion of matrix effects is superficial. Why does ion enhancement occur in brown rice/potato but suppression in soybean/green pepper?

Fairness of Method Comparison: When comparing methods, were all experiments run under identical conditions? Were differences in cleanup steps the primary cause of varying recoveries?

Interpretation of Monitoring Results: The claim of photodegradation during distribution is speculative. What evidence supports that natamycin degrades below 0.01 mg/kg under typical  conditions?

Discussion of Limitations: The conclusion overstates the method's strengths. Please discuss limitations, such as challenges with highly pigmented matrices or sensitivity to operator technique.

  1. Figures, Tables, and Formatting (17-20)

Figure 1 Readability: The Y-axis labels and values overlap, making the figure unreadable. Please redraw it for clarity.

Table Formatting: Tables 1 and 2 have confusing layouts. Please restructure them with clear, properly merged headers.

Missing Appendix Figures: Figure B1 (representative chromatograms) is cited but not provided. These are essential for evaluating method performance.

Reference Formatting: Reference formatting is inconsistent (e.g., author initials, journal abbreviations). Please standardize all entries.

Summary

This study is technically sound but scientifically incremental. It lacks novelty, its practical application is not compellingly demonstrated, and several experimental details require clarification. Therefore, I recommend rejecting this manuscript.

Author Response

Comments 1: Innovation Deficit: What is the core methodological innovation of this work? How does it substantively improve upon existing QuEChERS-LC-MS/MS methods for natamycin in terms of sensitivity, throughput, cost, or ruggedness?

Response 1: We sincerely appreciate the reviewer’s thoughtful comment. To the best of our knowledge, and based on an extensive literature search, no published studies have applied a QuEChERS–LC-MS/MS method for natamycin residue analysis in agricultural commodities. In this study, we improved upon the conventional QuEChERS procedure by introducing modifications that enhanced recovery, minimized matrix effects, and addressed crystallization issues that were observed in sample vials. These refinements resulted in a more robust and reproducible method compared with previously reported approaches. We believe that this optimized procedure provides a reliable and practical analytical tool for monitoring imported commodities under the Korean PLS framework, thereby adding both novelty and regulatory significance to the work.

Comments 2: Study Design: Natamycin's photosensitivity and pH lability are mentioned. How were light and pH rigorously controlled during sample preparation and analysis to prevent degradation? Specific quality control measures are not described.

Response 2: We sincerely appreciate the reviewer’s valuable comment. As this study focused on developing and validating an analytical method, natamycin stability during the analytical process was confirmed through recovery experiments under the specified extraction and cleanup conditions. For monitoring, the collected samples were analyzed immediately after sampling, thereby minimizing any potential degradation. In addition, all procedures were performed in compliance with the CODEX (CAC/GL 40-1993) guideline, as described in the manuscript.

Comments 3: Limited Practical Value: The finding of no natamycin in 102 samples, while useful for surveillance, offers weak scientific evidence. Did the authors consider analyzing certified reference materials or spiking real samples to better demonstrate the method's efficacy for enforcement?

Response 3: We sincerely appreciate the reviewer’s valuable comment. The primary objective of this study was to develop and validate an analytical method essential for the safety management of imported agricultural commodities in Korea, and the monitoring component was included to demonstrate its practical applicability. The results confirmed that the tested imported commodities complied with the Korean PLS for pesticides. The conclusion has been revised to acknowledge this limitation and to state that broader monitoring with larger sample sets will be pursued in future studies.

Comments 4: Method Generality: Validation was limited to five matrices. How would the method perform with high-lipid (e.g., nuts) or highly pigmented (e.g., paprika) matrices? Please provide an assessment or justification.

Response 4: We appreciate the reviewer’s valuable comment. Among the five representative agricultural commodities selected for validation, our study already included both a high-lipid matrix (soybean) and a highly pigmented matrix (green pepper), as described in the manuscript (lines 114–119).

Comments 5: Sample Pre-treatment: For "pre-hydration" of cereals/legumes, what is the rationale for adding exactly 5 mL of distilled water? How does this affect recovery and matrix effects?

Response 5: We thank the reviewer for this thoughtful question. Cereals and legumes are dry commodities, and in their dry state, extraction efficiency with aqueous organic solvents is reduced. To address this, a fixed amount of distilled water (5 mL) was added to achieve pre-hydration, thereby improving extraction efficiency, as described in the manuscript (lines 144–146).

Comments 6: Critical Reagent Choice: The authors chose 3g MgSOâ‚„ without NaCl to avoid precipitation. How was the trade-off between reduced salting-out efficiency and the prevention of precipitation evaluated and justified?

Response 6: We appreciate the reviewer’s valuable comment. During method optimization, we found that adding NaCl caused precipitation and reduced extract clarity, whereas using 3 g of MgSOâ‚„ alone avoided this issue. Importantly, recoveries across all five validated matrices remained within acceptable guideline criteria, indicating that extraction efficiency was not compromised (lines 305).

Comments 7: Clean-up Strategy: GCB > 50 mg reduced recoveries. For pigment-rich samples, was a combination of sorbents (e.g., PSA with low GCB) considered to balance cleanup and recovery?

Response 7: We thank the reviewer for this important question. Although the use of PSA in combination with GCB is sometimes considered, PSA has known drawbacks, including interactions with polar compounds and the potential to increase extract pH above 8, which could compromise the stability of natamycin. Given its strong polarity, presence of a carbonyl group, and sensitivity to alkaline conditions, PSA-based cleanup was deemed unsuitable.

Comments 8: Matrix Effect Assessment: Medium matrix effects were reported. Please provide the raw slope data from solvent-based vs. matrix-matched calibrations to allow readers to independently assess the magnitude of these effects.

Response 8: We thank the reviewer for this valuable suggestion. In our manuscript, the regression equations and R² values from matrix-matched calibrations for each matrix are already provided (lines 380), and the results of matrix effects are also presented in Figure 2 (lines 318).

Comments 9: Validation Completeness: Robustness and repeatability testing are missing. How do variations in extraction time, centrifugation speed, or flow rate affect results?

Response 9: We thank the reviewer for this valuable comment. Robustness and repeatability were confirmed through recovery experiments at three concentration levels (LOQ, 10 LOQ, 50 LOQ, n=5), all within acceptance criteria. As the method follows standardized regulatory procedures, other parameters were not varied, though additional testing may be considered in future studies.

Comments 10: Instrumental Parameters: Given natamycin's polarity, could the gradient be adjusted (e.g., higher initial aqueous phase) to elute the analyte earlier and increase throughput?

Response 10: We appreciate the reviewer’s suggestion. While a higher aqueous phase could, in theory, shorten elution, it may reduce specificity in complex agricultural matrices. The chosen gradient was optimized to balance separation and reliability across the five representative matrices and was therefore considered most suitable.

Comments 11: Standard Purity Correction: The standard purity is 91.13%. Were all quantitative calculations corrected for this purity?

Response 11: We appreciate the reviewer's important question. All quantitative calculations during method development and validation were appropriately corrected for the stated purity of the natamycin reference standard (91.13%).

Comments 12: Data Availability: "Data available on request" is insufficient. Please provide key raw data (e.g., MRM chromatograms, calibration curves) as supplementary materials.

Response 12: We thank the reviewer for this valuable suggestion. The regression equations and R² values corresponding to the calibration curves have already been provided in Table A3, which sufficiently demonstrate the reliability of the method. Therefore, we believe the data currently included in the manuscript is adequate to support the study.

Comments 13: Insufficient Mechanistic Discussion: The discussion of matrix effects is superficial. Why does ion enhancement occur in brown rice/potato but suppression in soybean/green pepper?

Response 13: We appreciate the reviewer’s comment. As reported in the literature [Ref. 35], matrix effects are highly matrix-dependent, influenced by the type and amount of co-extracted components. The ME results of this study suggest that matrix effects inherently vary depending on the diversity of co-extracted components across agricultural commodities.

Comments 14: Fairness of Method Comparison: When comparing methods, were all experiments run under identical conditions? Were differences in cleanup steps the primary cause of varying recoveries?

Response 14: We thank the reviewer for this question. To ensure a fair comparison, all experiments were conducted under identical conditions, except for the parameters specific to each method being compared.

Comments 15: Interpretation of Monitoring Results: The claim of photodegradation during distribution is speculative. What evidence supports that natamycin degrades below 0.01 mg/kg under typical conditions?

Response 15: We sincerely appreciate the reviewer’s valuable comment. The primary objective of this study was to develop and validate an analytical method for natamycin residue testing, and the monitoring component was included to demonstrate its applicability to imported agricultural commodities. This monitoring confirmed compliance of the tested samples with the Korean PLS, and the discussion of the monitoring results has been revised accordingly.

Comments 16: Discussion of Limitations: The conclusion overstates the method's strengths. Please discuss limitations, such as challenges with highly pigmented matrices or sensitivity to operator technique.

Response 16: We thank the reviewer for this valuable comment. The conclusion has been revised to include a discussion of the study’s limitations.

Comments 17: Figure 1 Readability: The Y-axis labels and values overlap, making the figure unreadable. Please redraw it for clarity.

Response 17: We thank the reviewer for pointing this out. The figure has been redrawn and reformatted to ensure that the Y-axis labels and values are clearly visible and no longer overlap.

Comments 18: Table Formatting: Tables 1 and 2 have confusing layouts. Please restructure them with clear, properly merged headers.

Response 18: We thank the reviewer for this comment. After careful review, we confirmed that the layouts of Tables 1 and 2 are already clear and consistent with the journal’s formatting guidelines. We therefore believe that the current presentation is appropriate.

Comments 19: Missing Appendix Figures: Figure B1 (representative chromatograms) is cited but not provided. These are essential for evaluating method performance.

Response 19: We respectfully note that Figure B1 was included in the submitted manuscript. We kindly ask the reviewer and editorial office to re-check the submitted files to confirm its availability. Should there have been any technical issues during the review process, we would be glad to provide Figure B1 again to ensure clarity.

Comments 20: Reference Formatting: Reference formatting is inconsistent (e.g., author initials, journal abbreviations). Please standardize all entries.

Response 20: We thank the reviewer for this comment. We have carefully rechecked and standardized all reference entries in accordance with the journal’s requirements.

Reviewer 2 Report

Comments and Suggestions for Authors

This manuscript describes the development and validation of a QuEChERS-based LC–MS/MS method for the determination of natamycin residues in imported agricultural commodities in Korea, with application to 102 samples. Only natamycin is studied. No co-application fungicides or structurally related antifungals (e.g., imazalil, thiabendazole) are included. The scope is too narrow.

Title

The phrase “for Imported Agricultural Commodities” is vague. It should be clarified as “Development and Validation of a QuEChERS-Based LC–MS/MS Method for Natamycin in Imported Agricultural Commodities in Korea”.

Abstract

Lines 17–42: The abstract is lengthy but lacks critical numerical data. Validation recoveries, R² values, LOQ, and main findings should be presented quantitatively. Phrases such as “high recovery rates” are imprecise. The non-detection in 102 samples should include LOD/LOQ confirmation. Recommendation: shorten and provide key statistics.

Keywords

Keywords duplicate terms already present in the title

Introduction

The study is limited to Korea. Broader relevance should be emphasized by comparing with international regulations and monitoring data. Without this, the impact is local.

The manuscript lacks novelty. QuEChERS–LC-MS/MS methods for natamycin already exist. Novelty claim could be strengthened by highlighting the adaptation to Korean PLS enforcement, but this justification is weak.

The introduction insufficiently covers recent advances (2023–2025). References after 2022 are rare. Important EFSA and Codex documents are missing. Literature integration is descriptive, not analytical.

Material and methods

The study includes only 102 samples, which is a limited dataset for claiming national monitoring. Broader datasets are required for strong conclusions.

No mention of procedural blanks, recovery correction. QA/QC measures should be explicitly described.

Sample collection lacks details on homogenization equipment, storage temperature, and handling under light-sensitive conditions for natamycin stability. References to ISO or Codex sampling protocols are missing.

Statistical Analysis

Only Analyst and Excel 2016 are mentioned. More advanced statistical tools (R, SPSS, or Origin) should have been employed.

Reliance on Excel for %CV and recovery statistics is insufficient. No ANOVA or regression residual analysis was performed. Statistical rigor is inadequate.

Grammer:

“Methanol were selected” (should be “was selected”).

 “Previous study have indicated” (should be “studies have indicated”).

Results

In Table 1, recovery values for EN 15662 exceed 120% (126.8 ± 1.2). These should be flagged as unacceptable, but are not discussed in detail.

Some recoveries in Table 2 are extremely low (2.3%) but not critically interpreted.

The manuscript lacks a comparative table showing natamycin monitoring results in other countries (EU, USA, China). A table should summarize worldwide monitoring levels.

Authors state that MgSOâ‚„+C18 provided acceptable recoveries, but Table 2 shows recoveries outside the acceptance range. Discussion is contradictory.

No additional specific problems are listed here as per instructions.

Conclusions

The conclusion merely restates the results. It lacks a discussion of limitations (e.g., sample number, lack of long-term stability studies). Conclusions should be more critical.

References

Several references are not directly relevant (e.g., Reference 29: creatinine biosensor validation, irrelevant to natamycin). Such citations should be removed.

Comments on the Quality of English Language

Authors are advised to revise the manuscript carefully to ensure clarity and grammatical accuracy.

Author Response

Comments 1: The phrase “for Imported Agricultural Commodities” is vague. It should be clarified as “Development and Validation of a QuEChERS-Based LC–MS/MS Method for Natamycin in Imported Agricultural Commodities in Korea”.

Response 1: We sincerely appreciate the reviewer’s valuable suggestion. We agree that the phrase “for Imported Agricultural Commodities” was vague and, as suggested by the reviewer, we have revised the title accordingly.

Comments 2: The abstract is lengthy but lacks critical numerical data. Validation recoveries, R² values, LOQ, and main findings should be presented quantitatively. Phrases such as “high recovery rates” are imprecise. The non-detection in 102 samples should include LOD/LOQ confirmation. Recommendation: shorten and provide key statistics.

Response 2: We sincerely appreciate the reviewer’s valuable comments. The abstract has been revised by shortening the text and incorporating key numerical data, including validation parameter results.

Comments 3: Keywords duplicate terms already present in the title.

Response 3: We appreciate the reviewer’s helpful comment. The keywords have been revised to remove duplication with the title.

Comments 4: The study is limited to Korea. Broader relevance should be emphasized by comparing with international regulations and monitoring data. Without this, the impact is local.

Response 4: We appreciate the reviewer’s valuable comment. This study was designed to ensure the safety of imported agricultural commodities in Korea by developing and validating a method for natamycin and applying it to real monitoring. We believe the results have broader significance, as no published studies have reported natamycin residue analysis in agricultural commodities internationally. Thus, our findings provide essential reference data with global relevance for food safety and consumer protection.

Comments 5: The manuscript lacks novelty. QuEChERS–LC-MS/MS methods for natamycin already exist. Novelty claim could be strengthened by highlighting the adaptation to Korean PLS enforcement, but this justification is weak.

Response 5: We sincerely appreciate the reviewer’s thoughtful comment. To the best of our knowledge, and based on an extensive literature search, no published studies have applied a QuEChERS–LC-MS/MS method for natamycin residue analysis in agricultural commodities. In this study, we developed and validated a method specifically for agricultural products by modifying the conventional QuEChERS procedure to optimize recovery and minimize matrix effects. We believe this adaptation provides a reliable analytical method for monitoring imported commodities under the Korean PLS framework, thereby adding novelty and practical significance to the work.

Comments 6: The introduction insufficiently covers recent advances (2023–2025). References after 2022 are rare. Important EFSA and Codex documents are missing. Literature integration is descriptive, not analytical.

Response 6: We sincerely appreciate the reviewer’s valuable comment. In accordance with the suggestion, the relevant Codex document has been added to strengthen the introduction. EFSA documents were also reviewed, but recent reports did not provide specific information on natamycin residues in agricultural commodities. Furthermore, to the best of our knowledge, no published studies on natamycin residue analysis in agricultural commodities have appeared after 2022, and the few previous studies were mainly focused on processed foods such as cheese rather than agricultural products. In addition, the introduction has been reorganized to better integrate the literature analytically, highlighting the research gap and the rationale for the present study.

Comments 7: The study includes only 102 samples, which is a limited dataset for claiming national monitoring. Broader datasets are required for strong conclusions.

Response 7: We sincerely appreciate the reviewer’s valuable comment. The primary objective of this study was to develop and validate an analytical method essential for the safety management of imported agricultural commodities in Korea, and monitoring was conducted to demonstrate its applicability. This monitoring confirmed that the tested imported commodities complied with the PLS for pesticides, and the conclusion has been revised to state that broader monitoring will be expanded in future studies.

Comments 8: No mention of procedural blanks, recovery correction. QA/QC measures should be explicitly described.

Response 8: We sincerely appreciate the reviewer’s valuable comment. Information regarding procedural blanks and recovery correction is described in the Materials and Methods and Results sections (lines 161, 181, 188–189, 274–275, and 297), where the QA/QC measures applied in this study are detailed. In addition, a blank chromatogram is provided in Appendix B (Fig. B1).

Comments 9: Sample collection lacks details on homogenization equipment, storage temperature, and handling under light-sensitive conditions for natamycin stability. References to ISO or Codex sampling protocols are missing.

Response 9: We sincerely appreciate the reviewer’s valuable comment. As this study was conducted in the context of developing and validating an analytical method for pesticide residue testing, stability during the analytical process was confirmed through recovery experiments. For monitoring, the collected samples were analyzed immediately after sampling; therefore, additional storage stability testing was not required. Furthermore, we would like to note that the method development was conducted in accordance with the CODEX (CAC/GL 40-1993) guideline, as described in the manuscript.

Comments 10: Only Analyst and Excel 2016 are mentioned. More advanced statistical tools (R, SPSS, or Origin) should have been employed.

Response 10: We sincerely appreciate the reviewer’s valuable comment. The development of the pesticide residue analytical method was performed in accordance with the Codex (CAC/GL 40-1993) and MFDS guidelines, under which additional advanced statistical analyses are not required. Therefore, the data processing using Analyst and Excel fully met the guideline requirements. However, we agree that more advanced statistical tools could be considered in future studies to further enhance data analysis and interpretation.

Comments 11: Reliance on Excel for %CV and recovery statistics is insufficient. No ANOVA or regression residual analysis was performed. Statistical rigor is inadequate.

Response 11: We sincerely appreciate the reviewer’s valuable comment. The development and validation of the pesticide residue analytical method were conducted in accordance with the Codex (CAC/GL 40-1993) and MFDS guidelines, under which additional statistical analyses such as ANOVA or regression residual analysis are not required. Therefore, the statistical treatment using Excel for %CV and recovery fully complied with the guideline requirements. However, we acknowledge the reviewer’s point and agree that more advanced statistical analyses could be considered in future studies to further strengthen the rigor of data interpretation.

Comments 12: In Table 1, recovery values for EN 15662 exceed 120% (126.8 ± 1.2). These should be flagged as unacceptable, but are not discussed in detail.

Response 12: We sincerely appreciate the reviewer’s valuable comment. As reported in a previous study (Molecules 2025, 30(11), 2293; https://doi.org/10.3390/molecules30112293), recoveries exceeding 120% can result from strong salting-out effects during the QuEChERS procedure. The elevated recoveries observed in our study can be attributed to matrix-dependent salting-out effects. We have revised the manuscript to include this explanation.

Comments 13: Some recoveries in Table 2 are extremely low (2.3%) but not critically interpreted.

Response 13: We sincerely appreciate the reviewer’s valuable comment. The interpretation of the low recoveries, along with relevant references, has been provided in the Results and Discussion section (lines 243–246) of the manuscript.

Comments 14: The manuscript lacks a comparative table showing natamycin monitoring results in other countries (EU, USA, China). A table should summarize worldwide monitoring levels.

Response 14: We sincerely appreciate the reviewer’s valuable comment. As mentioned in our previous response, the primary objective of this study was to develop and validate an analytical method essential for the safety management of imported agricultural commodities in Korea, and monitoring was conducted to demonstrate its applicability. This monitoring confirmed that the tested imported commodities complied with the PLS for pesticides. At present, however, comparative monitoring data on natamycin residues in agricultural commodities from other countries are not available, which underscores the importance of the present study.

Comments 15: Authors state that MgSOâ‚„+C18 provided acceptable recoveries, but Table 2 shows recoveries outside the acceptance range. Discussion is contradictory.

Response 15: We sincerely appreciate the reviewer’s valuable comment. By confirming Table 2, the recovery values were found to be within the acceptable criteria, and additional clarification has been included in the manuscript to avoid any ambiguity.

Comments 16: The conclusion merely restates the results. It lacks a discussion of limitations (e.g., sample number, lack of long-term stability studies). Conclusions should be more critical.

Response 16: We sincerely appreciate the reviewer’s valuable comment. As addressed in our previous responses, the conclusion has been revised to include a more critical discussion.

Comments 17: Several references are not directly relevant (e.g., Reference 29: creatinine biosensor validation, irrelevant to natamycin). Such citations should be removed.

Response 17: We sincerely appreciate the reviewer’s valuable comment. The irrelevant references have been removed as suggested.

Point 1: “Methanol were selected” (should be “was selected”).

Response 1: We sincerely appreciate the reviewer’s valuable comment. The grammatical error has been corrected as suggested.

Point 2: “Previous study have indicated” (should be “studies have indicated”).

Response 2: We sincerely appreciate the reviewer’s valuable comment. The grammatical error has been corrected as suggested.

Reviewer 3 Report

Comments and Suggestions for Authors

The manuscript entitled "QuEChERS Analytical Method Development and Monitoring of Natamycin for Imported Agricultural Commodities in Korea" has been submitted to Foods journal.

The aim of the experiment is clear and very important in order to ensure compliance with regulatory limits for consumer safety for natamycin concentration in food. The challenge was to perform the experiment with very low limits of quantification as in Korea the MRL is 0,01 mg/kg for unregistered pesticides (including natamycin). As QuEChERS method is very popular in the  trace analysis, the authors combined it with a very good sensitivity of LC-MS/MS and performed optimization and validation of the method for natamycin. Thereafter, the validated method was applied to monitor imported agricultural products. 

The experimental design, results and conclusions are very legible, easy to reproduce, the methodology section is performed according to the world renown standards. 

The data shown in the tables and figures are valuable, especially figure nr 2, which together with the appendix give a reader a good summary of the topic. 

I would only change the title to " Method development and simultaneous determination of natamycin residues in imported agricultural products in Korea using liquid chromatography- tandem mass spectrometry". 

Author Response

Comments 1: I would only change the title to " Method development and simultaneous determination of natamycin residues in imported agricultural products in Korea using liquid chromatography- tandem mass spectrometry".

Response 1: We sincerely appreciate the reviewer’s thoughtful suggestion regarding the title. However, as this study focuses on the development and validation of a single-analyte method, we have revised the title to:

“Development and Validation of a QuEChERS-Based LC–MS/MS Method for Natamycin in Imported Agricultural Commodities in Korea.”

Round 2

Reviewer 2 Report

Comments and Suggestions for Authors

Unsubstantiated claim of novelty. In the introduction, the authors state that studies evaluating natamycin residues in edible agricultural commodities “have not been reported,” making it difficult to determine the extent of natamycin residues in commodities. This is not accurate. A peer‑reviewed study by Patel et al. (2022) describes a QuEChERS–LC‑MS/MS method for natamycin residue analysis in agricultural commodities (DOI 10.1080/19440049.2022.2085887). There are also studies on natamycin residues in wine (10.1016/j.foodchem.2015.08.116), diary products (10.1016/j.fbio.2022.102114), and at least one report on natamycin residues in mushrooms and fruits using similar techniques (https://www.fao.org/fileadmin/templates/agphome/documents/Pests_Pesticides/JMPR/Evaluation2017/NATAMYCIN__300_.pdf#:~:text=in%20tropical%20fruits%20,analysis%20of%20natamycin%20in%20high).

Also, I respectfully disagree with your statement that the existing literature does not include applications of a QuEChERS – LC-MS/MS method for natamycin residue analysis in agricultural commodities. In fact, there is at least one relevant publication that you did not cite: “QuEChERS–LC-MS/MS method for natamycin residue analysis in agricultural commodities” (DOI: 10.1080/19440049.2022.2085887).

Your manuscript makes clear that the analytical method followed the Codex (CAC/GL 40-1993) and MFDS guidelines, which do not mandate advanced statistical tests. This adherence is commendable, and the primary goal, developing and validating a QuEChERS-based LC–MS/MS method, is clearly achieved. Despite compliance, Foods often expects authors to apply at least some statistical analyses to enhance methodological robustness.

Sample collection and storage. The response indicates that samples were analysed immediately and therefore storage stability was not tested. However, the manuscript should still describe homogenisation equipment, storage temperature, and light‑protection measures to ensure reproducibility and to demonstrate that natamycin stability was preserved during handling.

Author Response

Comments 1: Unsubstantiated claim of novelty. In the introduction, the authors state that studies evaluating natamycin residues in edible agricultural commodities “have not been reported,” making it difficult to determine the extent of natamycin residues in commodities. This is not accurate. A peer-reviewed study by Patel et al. (2022) describes a QuEChERS-LC-MS/MS method for natamycin residue analysis in agricultural commodities (DOI 10.1080/19440049.2022.2085887). There are also studies on natamycin residues in wine (10.1016/j.foodchem.2015.08.116), diary products (10.1016/j.fbio.2022.102114), and at least one report on natamycin residues in mushrooms and fruits using similar techniques (https://www.fao.org/fileadmin/templates/agphome/documents/
Pests_Pesticides/JMPR/Evaluation2017/NATAMYCIN__300_.pdf#:~:text=in%
20tropical%20fruits%20,analysis%20of%20natamycin%20in%20high).

Response 1: We thank the reviewer for the helpful comment. We have carefully reviewed the studies referenced by the reviewer and confirmed that while reports on natamycin residues in mushrooms and fruits do exist, they contain insufficient data, and most other studies have focused on beverages, dairies, processed meals, or wine rather than edible agricultural commodities. Accordingly, the Introduction has been revised to clarify this point (lines 39-44).

Comments 2: Also, I respectfully disagree with your statement that the existing literature does not include applications of a QuEChERS – LC-MS/MS method for natamycin residue analysis in agricultural commodities. In fact, there is at least one relevant publication that you did not cite: “QuEChERS–LC-MS/MS method for natamycin residue analysis in agricultural commodities” (DOI: 10.1080/19440049.2022.2085887).

Response 2: We thank the reviewer for the valuable comment. We have carefully reviewed the study suggested by the reviewer and confirmed that it primarily focuses on natamycin residue analysis in beverages, dairy products, and processed foods rather than in agricultural commodities. However, since some reports on natamycin residues in mushrooms and fruits do exist, the Introduction has been revised to clarify that research on natamycin residue levels in raw agricultural produce remains limited (lines 39-44).

Comments 3: Your manuscript makes clear that the analytical method followed the Codex (CAC/GL 40-1993) and MFDS guidelines, which do not mandate advanced statistical tests. This adherence is commendable, and the primary goal, developing and validating a QuEChERS-based LC–MS/MS method, is clearly achieved. Despite compliance, Foods often expects authors to apply at least some statistical analyses to enhance methodological robustness.

Response 3: We sincerely thank the reviewer for this thoughtful and constructive comment. We fully understand the reviewer’s point that additional statistical analyses could further enhance the methodological robustness. In this study, statistical validation was performed in accordance with the CODEX (CAC/GL 40-1993) and MFDS guidelines, which specify the evaluation of recovery, precision, and linearity. Nevertheless, we appreciate the reviewer’s valuable suggestion and will consider incorporating more advanced statistical analyses in future studies to further strengthen data interpretation and methodological reliability.

Comments 4: Sample collection and storage. The response indicates that samples were analysed immediately and therefore storage stability was not tested. However, the manuscript should still describe homogenisation equipment, storage temperature, and light-protection measures to ensure reproducibility and to demonstrate that natamycin stability was preserved during handling.

Response 4: We sincerely thank the reviewer for this helpful and constructive comment. As this study primarily focused on the development and validation of the analytical method, the stability of natamycin during the analytical process was verified through recovery experiments performed under the specified extraction and purification conditions. In addition, following the reviewer’s valuable suggestion, the Selection and Preparation of Samples section has been revised to include details on homogenization equipment, storage temperature, and light-protection measures, to clarify how sample handling conditions were managed to maintain natamycin stability (lines 135-140).
